# Effects of Methylprednisolone on Ventilator-Free Days in Mechanically Ventilated Patients with Acute Respiratory Distress Syndrome and COVID-19: A Retrospective Study

**DOI:** 10.3390/jcm10040760

**Published:** 2021-02-14

**Authors:** Mohamed Badr, Bruno De Oliveira, Khaled Abdallah, Ashraf Nadeem, Yeldho Varghese, Dnyaseshwar Munde, Shameen Salam, Baraa Abduljawad, Khaled Saleh, Hussam Elkambergy, Ahmed Taha, Ahmed Bayrlee, Ali Wahla, Jamil Dibu, Rehan Haque, Fadi Hamed, Nadeem Rahman, Jihad Mallat

**Affiliations:** 1Critical Care Institute, Cleveland Clinic Abu Dhabi, Abu Dhabi 112412, United Arab Emirates; rashadicu@gmail.com (M.B.); deolivb@clevelandclinicabudhabi.ae (B.D.O.); Dr_khaled_Salah@windoslive.com (K.A.); ashmohnad@hotmail.com (A.N.); VargheY2@ClevelandClinicAbuDhabi.ae (Y.V.); MundeD@ClevelandClinicAbuDhabi.ae (D.M.); SalamS3@ClevelandClinicAbuDhabi.ae (S.S.); AbduljB@ClevelandClinicAbuDhabi.ae (B.A.); salehk@clevelandclinicabudhabi.ae (K.S.); ElkambH@ClevelandClinicAbuDhabi.ae (H.E.); TahaA2@ClevelandClinicAbuDhabi.ae (A.T.); BayrleA@ClevelandClinicAbuDhabi.ae (A.B.); wahlaa@clevelandclinicabudhabi.ae (A.W.); DibuJ@ClevelandClinicAbuDhabi.ae (J.D.); HaqueR@ClevelandClinicAbuDhabi.ae (R.H.); HamedF@ClevelandClinicAbuDhabi.ae (F.H.); RahmanN2@ClevelandClinicAbuDhabi.ae (N.R.); 2Cleveland Clinic Lerner College of Medicine of Case Western Reserve University, Cleveland, OH 44195, USA; 3Faculty of Medicine, Normandy University, UNICAEN, ED 497, 1400 Caen, France

**Keywords:** COVID-19, acute respiratory distress syndrome, methylprednisolone, mechanical ventilation, ventilator-free days, SARS-CoV-2 infection

## Abstract

*Objectives:* There are limited data regarding the efficacy of methylprednisolone in patients with acute respiratory distress syndrome (ARDS) due to coronavirus disease 2019 (COVID-19) requiring invasive mechanical ventilation. We aimed to determine whether methylprednisolone is associated with increases in the number of ventilator-free days (VFDs) among these patients. *Design:* Retrospective single-center study. *Setting:* Intensive care unit. *Patients:* All patients with ARDS due to confirmed SARS-CoV-2 infection and requiring invasive mechanical ventilation between 1 March and 29 May 2020 were included. *Interventions:* None. *Measurements and Main Results:* The primary outcome was ventilator-free days (VFDs) for the first 28 days. Defined as being alive and free from mechanical ventilation. The primary outcome was analyzed with competing-risks regression based on Fine and Gray’s proportional sub hazards model. Death before day 28 was considered to be the competing event. A total of 77 patients met the inclusion criteria. Thirty-two patients (41.6%) received methylprednisolone. The median dose was 1 mg·kg^−1^ (IQR: 1–1.3 mg·kg^−1^) and median duration for 5 days (IQR: 5–7 days). Patients who received methylprednisolone had a mean 18.8 VFDs (95% CI, 16.6–20.9) during the first 28 days vs. 14.2 VFDs (95% CI, 12.6–16.7) in patients who did not receive methylprednisolone (difference, 4.61, 95% CI, 1.10–8.12, *p* = 0.001). In the multivariable competing-risks regression analysis and after adjusting for potential confounders (ventilator settings, prone position, organ failure support, severity of the disease, tocilizumab, and inflammatory markers), methylprednisolone was independently associated with a higher number of VFDs (subhazards ratio: 0.10, 95% CI: 0.02–0.45, *p* = 0.003). Hospital mortality did not differ between the two groups (31.2% vs. 28.9%, *p* = 0.82). Hospital length of stay was significantly shorter in the methylprednisolone group (24 days [IQR: 15–41 days] vs. 37 days [IQR: 23–52 days], *p* = 0.046). The incidence of positive blood cultures was higher in patients who received methylprednisolone (37.5% vs. 17.8%, *p* = 0.052). However, 81% of patients who received methylprednisolone also received tocilizumab. The number of days with hyperglycemia was similar in the two groups. *Conclusions:* Methylprednisolone was independently associated with increased VFDs and shortened hospital length of stay. The combination of methylprednisolone and tocilizumab was associated with a higher rate of positive blood cultures. Further trials are needed to evaluate the benefits and safety of methylprednisolone in moderate or severe COVID-19 ARDS.

## 1. Introduction

Severe acute respiratory syndrome coronavirus-2 (SAR-CoV-2) responsible for the coronavirus disease 2019 (COVID-19) has hit the world as a global pandemic at an unparalleled scale, causing considerable morbidity and mortality [1,2,3,4]. Most people with COVID-19 have only mild or uncomplicated disease. However, up to 12% of hospitalized patients can progress to critical illness with acute respiratory distress syndrome (ARDS) requiring invasive mechanical ventilation [5,6,7].

The histological features of COVID-19 ARDS are dominated by diffuse alveolar damage, inflammatory cell infiltration, and microvascular thrombosis [8,9]. Patients with severe COVID-19 present nonspecific hyperinflammatory responses with a markedly elevated number of proinflammatory cytokines and chemokines [10,11,12]. This excessive and deleterious host immune response is thought to contribute to multi-organ failure in these patients. Corticosteroids might mitigate this exacerbated inflammatory response by inhibiting the expression of proinflammatory cytokines [13,14]. Therefore, there is a significant interest in using corticosteroids to treat severe COVID-19 patients.

Findings from the recently published RECOVERY trial demonstrated that the use of dexamethasone reduced mortality, especially in the subgroup of patients requiring invasive mechanical ventilation [15]. However, the severity of hypoxemia, ventilator settings (tidal volume, plateau pressure, driving pressure), and other types of organ support (vasopressors use, prone position, etc.) were not reported in these patients but are associated with the outcome [16]. Although a recent prospective meta-analysis of individual data from seven randomized controlled trials of severe COVID-19 patients showed that systemic corticosteroids were associated with lower all-cause mortality. Most of the included patients did not receive invasive mechanical ventilation [17]. Only one randomized controlled trial that focused on moderate or severe ARDS patients with COVID-19 requiring invasive mechanical ventilation, which was stopped prematurely, reported that dexamethasone was associated with a significant increase in the number of ventilator-free days (VFDs) [18]. 

Methylprednisolone has lower potent anti-inflammatory effects and shorter plasma half-time than dexamethasone. To the best of our knowledge, there is only one retrospective published study that investigated the benefit of methylprednisolone in critically ill, mechanically ventilated COVID-19 patients [19]. In that study, the use of methylprednisolone was associated with increased VFDs. However, no adjustment for ventilator settings was reported in that study. Therefore, there is limited evidence for the efficacy of methylprednisolone in these patients. Therefore, our study aimed to evaluate methylprednisolone’s efficacy in ARDS patients requiring invasive mechanical ventilation due to COVID-19. The hypothesis was that methylprednisolone would increase the number of days alive and free from mechanical ventilation during the first 28 days after adjustment for potential confounders, including ventilator settings.

## 2. Methods

This retrospective study was approved by the institutional Ethics Committee of Cleveland Clinic Abu Dhabi (A-2020-055) and waived the need for informed consent due to the retrospective nature of the study.

All adult patients (age ≥ 18 years) admitted to our intensive care unit (ICU) between March 1st and May 29th, 2020, with confirmed SARS-CoV-2 infection (virus detected by a real-time reverse-transcriptase–polymerase-chain-reaction assay of a nasopharyngeal sample) and ARDS, according to the Berlin definition [20], requiring intubation and invasive mechanical ventilation, were included in this study. 

### 2.1. Outcome Measures

The primary outcome was VFDs during the first 28 days, defined as the number of days alive and free from mechanical ventilation for at least 48 consecutive hours [21]. Patients discharged from the hospital before 28 days were considered alive and free from mechanical ventilation at 28 days. Non-survivors at day 28 were considered to have no VFDs. For patients who died, the number of VFDs was 0. For patients who were alive, the VFDs were the days they did not require mechanical ventilation. For patients who required mechanical ventilation for more than 28 days, the number of VFDs was 0. 

Prespecified secondary outcomes were all-cause hospital mortality, ICU length of stay, hospital length of stay, and mechanical ventilation duration for 28 days.

We also assessed safety outcomes, including all positive cultures (blood, sputum, and urine) and hyperglycemia, defined as days with a blood glucose ≥ 10 mmol·L^−1^ for the first 28 days. 

### 2.2. Clinical and Laboratory Data

Data on baseline characteristics including demographics, physiological variables, the presence of medical comorbidities, SOFA and SAPS II score, and laboratory values including oxygenation parameters, full blood count, coagulation parameters, and inflammatory markers (C-reactive protein, interleukin 6, and ferritin) were collected on admission to the ICU. Ventilatory variables including plateau pressure (Pplat), total positive end-expiratory pressure (PEEP), tidal volume (Vt) driving pressure (Pplat-PEEP), and static respiratory compliance [(Vt/(Pplat-PEEP)] were also captured. Use of neuromuscular blocking agents, prone positioning, vasopressors, renal replacement therapy, anti-viral treatments, tocilizumab, and methylprednisolone were collected. The decision of using anti-viral tocilizumab and methylprednisolone administrations were at the discretion of the treating physician. The time from symptoms onset to methylprednisolone administration and dosing and duration of methylprednisolone were calculated. Patients were categorized according to whether they received or did not receive methylprednisolone during the ICU stay.

### 2.3. Statistical Analysis

Normality of data distribution was assessed using the Shapiro–Wilk test and by visually checking each variable’s distribution (histogram). Data are expressed as mean ± SD when normally distributed or as median [IQR] when non-normally distributed. Proportions were used as descriptive statistics for categorical variables. Comparisons of values between independent groups were performed by the 2-tailed Student *t*-test or the Mann–Whitney *U* test, as appropriate. Analysis of the discrete data was performed by *χ*^2^ test or the Fisher exact test when the numbers were small. There were missing data (missing at random) for interleukin 6 (3.6%), ferritin (1.8%), D-dimer (1%), fibrinogen (8.2%), APTT (9.1%), and INR (9.1%) that were imputed using multiple imputations with 50 imputed datasets.

The primary endpoint (VFDs) was evaluated with competing-risks regression based on Fine and Gray’s proportional subhazards model. Death before day 28 was considered to be the competing event, and time-to-event analysis was right-censored at 28 days. Adjusted competing-risks regression models were fitted to identify risk factors that were independently associated with VFDs, using clinically likely confounders including age, body mass index, comorbidities, SOFA II score, PaO_2_/FiO_2_ ratio, renal replacement therapy, use of vasopressors, methylprednisolone, time from symptoms onset to methylprednisolone administration, tocilizumab, C-reactive protein, interleukin 6, D-dimer, Vt, PEEP, Pplat, and driving pressure. Variables associated with VFDs (*p* < 0.1) in univariate analysis were also included in the adjusted competing-risks regression model. The potential problem of co-linearity was evaluated using Spearman or Pearson correlation coefficient before running the analysis. Sub-hazard ratios (SHRs) and 95% confidence intervals were summarized. 

A value of *p* < 0.05 was considered statistically significant, and all reported p values are two-sided. Statistical analyses were performed using Stata/SE 14.2 software for Windows (Stata Corp LLC, TX 77845, USA).

## 3. Results

### 3.1. Study Population

From March 1st to May 29th, 2020, 110 adult patients with ARDS caused by COVID-19 infection were admitted to the ICU. Among them, 77 required invasive mechanical ventilation and were included in this study (Figure 1). The main characteristics of the cohort are summarized in Table 1 and Table 2. The median age among all patients was 50 years (IQR: 41-5 years) and 72 years (93.5%) were men. Among the patients, 49 (63.4%) had at least one comorbidity, 50 (64.9%) were placed in a prone position, 65 (84.2%) required vasopressor support, and 24 (31.2%) received renal replacement therapy. The median time from symptoms onset to ICU admission was 5 days (IQR: 4–7 days). Thirty-two patients (41.6%) received methylprednisolone. The median methylprednisolone dose was 1 mg·kg^−1^ (IQR: 1–1.3 mg·kg^−1^) and median duration of 5 days (IQR: 5–7 days).

The median time from symptoms onset to methylprednisolone administration was 6 days (IQR: 5–11 days), and, from ICU admission to methylprednisolone infusion, was 0 days (IQR: 0–1 days). Patients’ characteristics (age, gender, and BMI), comorbidity, severity scores, and time from symptoms onset to ICU admission or intubation were not significantly different between the two groups (Table 1). Regarding laboratory data on ICU admission, only C-reactive protein and interleukin 6 were significantly lower in the methylprednisolone group than in the non-methylprednisolone group (Table 1). The treatments received in ICU (renal replacement therapy, antiviral drugs, tocilizumab, neuromuscular blockers, and prone position) did not differ significantly between the two groups except for the use of the vasopressor, which was significantly higher in the non-methylprednisolone group (Table 2).

### 3.2. Primary Clinical Outcome

In univariate analysis (without adjustment), methylprednisolone was associated with longer VFDs at day-28 (SHR = 0.46 [95%CI:0.25–0.85], *p* = 0.013) (Appendix A). In addition, age, chronic kidney disease, creatinine, procalcitonin, arterial oxygen saturation, lactate levels, vasopressor use, prone position, and neuromuscular blocker agents were found to be associated (*p* < 0.1) with VFDs (Appendix A). The mean number of days alive and free from mechanical ventilation during the first 28 days was significantly higher in the methylprednisolone group than in the non-methylprednisolone group (18.8, 95% CI, 16.6–20.9 days vs. 14.2, 95% CI, 12.6–16.7 days, difference, 4.61, 95% CI, 1.10–8.12, *p* = 0.001).

In the multivariable competing-risks regression analysis, after adjusting for the clinically known confounding variables (mentioned in the methods) along with those who were found associated with VFDs in univariate analysis (Appendix A), exposure to methylprednisolone was still significantly associated with longer VFDs at day-28 (SHR = 0.10 [95% CI:0.02–0.45], *p* = 0.003) (Table 3 and Figure 2). Creatinine level instead of chronic kidney disease was entered in the model to avoid collinearity.

### 3.3. Secondary Clinical Outcomes

There were no significant differences between methylprednisolone and non-methylprednisolone groups regarding the hospital mortality rate (31.2% vs. 28.9%, *p* = 0.82, respectively), median of ICU length of stay (15.5 days [IQR: 9.5–28 days] vs. 19 days [IQR: 15–37 days], *p* = 0.09, respectively), and median duration of mechanical ventilation (14.0 days [IQR: 6.5–23.5 days] vs. 17.0 days [IQR: 10.0–28.0 days], *p* = 0.08, respectively). However, the hospital length of stay was significantly shorter in the methylprednisolone group than in the non-methylprednisolone group (24 days [IQR: 15–41 days] for 37 days [IQR: 23–52 days], *p* = 0.046). 

Regarding safety outcomes, there were no significant differences between the two groups regarding the proportions of positive sputum cultures (60% vs. 57.8%, *p* = 0.85, respectively) and positive urine cultures (25% vs. 32%, *p* = 0.75, respectively). However, there was a higher proportion of positive blood cultures in patients who received methylprednisolone compared to those who did not (37.5% vs. 17.8%, *p* = 0.052, respectively). Nevertheless, 81% of patients who received methylprednisolone also received tocilizumab (Table 2). Days with blood glucose values ≥ 10 mmol·L^−1^ over the first 28 days were similar in the non-methylprednisolone group than in the methylprednisolone group (6.5 days [IQR: 2.5–14.5 days] vs. 8 days [IQR: 1–16 days], *p* = 0.98, respectively).

## 4. Discussion

In this study of COVID-19 ARDS, methylprednisolone treatment was independently associated with a longer number of days alive and free from mechanical ventilation during the first 28 days. Patients who received methylprednisolone also had a shorter hospital length of stay. However, methylprednisolone did not result in a decreased hospital mortality rate. In addition, methylprednisolone was associated with a higher proportion of positive blood cultures. 

Several observational trials assessed corticosteroids’ role for non-COVID-19 viral ARDS with conflicting results regarding their benefit and safety [22,23,24,25]. In critically ill patients with Middle East Respiratory Syndrome (MERS), corticosteroids were associated with an increased viral load without affecting mortality [22]. In addition, corticosteroids delayed viral clearance in patients with SARS-CoV [23]. A meta-analysis found higher mortality among patients with influenza treated with corticosteroids [24]. Recently, in hospitalized patients with COVID-19 pneumonia, the RECOVERY trial demonstrated that dexamethasone 6 mg daily for 10 days reduced the 28-day mortality, especially among patients receiving invasive mechanical ventilation [15]. However, incomplete information about some potential confounders (organ support, ventilator settings) related to the outcome may have caused an imbalance between the treated and control groups [16]. In the recent prospective meta-analysis of clinical trials of critically ill patients with COVID-19, administration of corticosteroids was associated with a decrease in 28-day all-cause mortality [17]. However, most of the trials included patients who did not require invasive mechanical ventilation. In addition, association with mortality was observed for patients who received dexamethasone (3 trials), but not for hydrocortisone (3 trials) or methylprednisolone (1 trial). Reports on the efficacy of methylprednisolone in ARDS patients with COVID-19 are conflicting [26,27]. In a randomized controlled trial that included patients with COVID-19 pneumonia, methylprednisolone treatment was not associated with improved mortality [26]. In a recent observational study, methylprednisolone was associated with a reduced risk of death within 60 days in COVID-19 patients with ARDS [27]. However, most of these studies did not focus on mechanically ventilated patients. Only one observational study investigated methylprednisolone’s efficacy in ARDS patients with COVID-19 requiring invasive mechanical ventilation [19]. The authors found that methylprednisolone use (1 mg·kg^−1^ daily for four to six days) was independently associated with higher VFDs. However, this study did not adjust for tidal volume, plateau pressure, driving pressure, use of neuromuscular blocker agents, and prone position, which are all well-known to affect the outcome [28,29,30,31]. Furthermore, the authors used a linear regression analysis to identify factors associated with VFDs, which is not appropriate to analyze censored and non-normally distributed variables like VFDs’ variables [21]. Therefore, these findings should be interpreted with caution.

The VFDs’ criterion was chosen as the primary outcome because it encompasses both mortality and ventilation duration in surviving patients [21]. We demonstrated that methylprednisolone use was independently associated with longer VFDs after adjusting for several potential confounders in competing-risks regression analysis (Table 3 and Figure 2). Our results are in agreement with those of a recent RCT (CoDEX trial) that included mechanically ventilated ARDS patients with COVID-19, where the authors found that dexamethasone treatment compared with standard of care alone resulted in a higher number of days alive and free from mechanical ventilation [18]. However, in the CoDEX trial, dexamethasone was used at a high dose (20 mg daily for five days, and 10 mg daily for the other five days). In our study, methylprednisolone was used at a lower dose (median dose of 1mg·kg-1 daily for a median duration of 5 days), consistent with the previous publications on COVID-19 patients with methylprednisolone treatment [19,26,27]. However, the optimal dose and duration of methylprednisolone treatment in patients with COVID-19 ARDS remain unknown and need further investigations.

We did not observe a difference in hospital mortality between methylprednisolone and non-methylprednisolone groups. However, our mortality rate (30%) is much lower than it was reported in the RECOVERY trial (41.4% in the usual care group for patients receiving invasive mechanical ventilation) [15], and in the CoDEX trial (61.5% in the standard care arm) [18], even though the PaO_2_/FiO_2_ ratio in our study was much lower (mean 104 mmHg) than in the CoDEX trial (mean 131). The discrepancy might be explained by the higher rate of prone position in our study (65%) compared to the CoDEX trial (22%) [18]. Our mortality rate is consistent with that observed (32%) recently from a large cohort of mechanically ventilated ARDS patients with COVID-19 (742 patients), in which 75% of patients received a prone position [3].

We observed a higher rate of positive blood cultures in patients who received methylprednisolone. This finding is in disagreement with previous studies that did not report an increase in the incidence of infectious complications among ARDS COVID-19 patients who received corticosteroids [18,19]. The higher rate of positive blood cultures in our study might be explained by the fact that 81% of patients who received methylprednisolone also received tocilizumab (Table 2). Thus, the combination of two immunosuppressive treatments might increase the likelihood of new infections. However, the higher rate of positive blood cultures did not impact the mortality or hospital length of stay. Consistent with previous reports [19], we did not observe a difference in the number of days with hyperglycemia (> 10 mmol·L^−1^) by day 28 between the two groups. 

Our results are of clinical importance and add significant data to the existing literature on methylprednisolone in ARDS COVID-19 mechanically ventilated patients. We used appropriate statistical analysis (competing-risks regression) and we adjusted for many potential confounders (ventilator settings, organ failure support, the severity of the disease, etc.). Methylprednisolone was associated with increases in the number of days alive and free from mechanical ventilation by 4.6 days and shortening hospital length of stay. These findings are relevant in the context of a pandemic where widely available and inexpensive treatment might alleviate the burden on the health care system.

Our study has some important limitations. First, it is a single-center retrospective study conducted at a quaternary care facility in the Middle East. Thus, management and outcomes do not necessarily reflect those at other centers. Second, despite multivariable analysis and adjustment for potential confounders, we cannot rule out bias selection or residual confounding. Thus, no causation can be inferred from our findings. Third, the majority of our patients were treated with tocilizumab (87%). Even though methylprednisolone’s efficacy was shown to be independent of tocilizumab use, our findings might not be generalizable to patients who are not receiving this treatment. Fourth, due to our small sample size, our study was not powered to detect a significant difference in mortality between the two groups.

## 5. Conclusions

Methylprednisolone was independently associated with the increase in the number of days alive and free of mechanical ventilation over 28 days in mechanically ventilated ARDS patients with COVID-19. Additionally, methylprednisolone resulted in a shortened hospital length of stay. However, the combination of methylprednisolone and tocilizumab was associated with a higher rate of positive blood cultures. Further trials are needed to evaluate the benefits and safety of methylprednisolone in moderate or severe COVID-19 ARDS. 

## Figures and Tables

**Figure 1 jcm-10-00760-f001:**
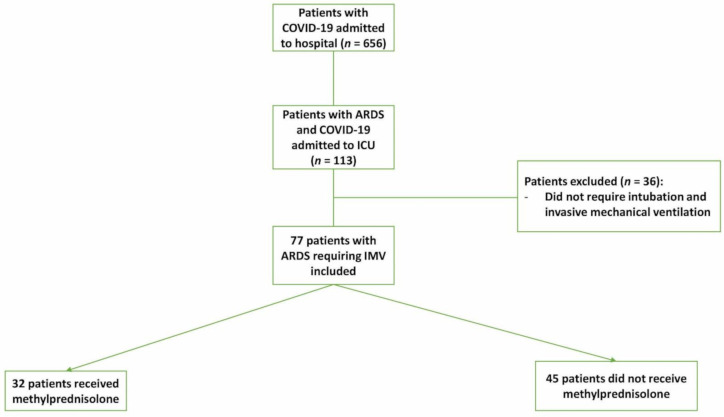
Flow chart of COVID-19 patients admitted to the intensive care unit. ARDS: acute respiratory distress syndrome. IMV: invasive mechanical ventilation.

**Figure 2 jcm-10-00760-f002:**
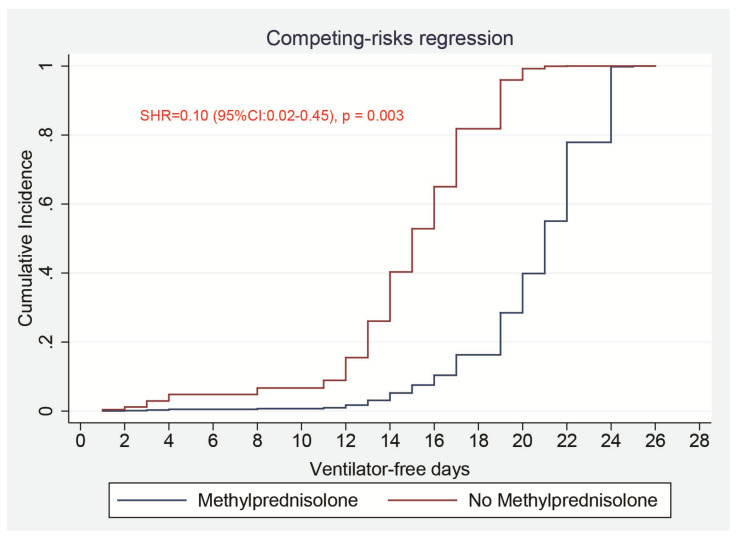
Comparisons of a cumulative incidence function of patients who were receiving invasive mechanical ventilation for 28 days between patients who received methylprednisolone and those who did not. SHR: subhazards ratio.

**Table 1 jcm-10-00760-t001:** Comparisons of baseline characteristics, laboratory data, and treatments during the intensive care unit (ICU) stay between Methylprednisolone and non-Methylprednisolone groups.

Variables	All Patients (*n* = 77)	Methylprednisolone (*n* = 32)	Non-Methylprednisolone (*n* = 45)	*p*-Value
Age, year	50 (41–59)	49 (41–55)	51 (41–61)	0.52
Male, *n* (%)	72 (93.5)	30 (93.7)	42 (93.3)	1.00
Body mass index, kg·m^−2^	25.8 (23.1–29.0)	25.8 (22.2–29.7)	25.9 (23.8–28.0)	0.96
SOFA score	7 (4–10)	7 (4–9)	7 (4–10)	0.80
SAPS II score	36 (27–48)	38 (28–49)	35 (27–46)	0.67
Patients with at least one comorbidity, *n* (%)	49 (63.4)	21 (65.6)	28 (62.2)	0.76
Comorbidities distribution, *n* (%)				
Diabetes mellitus	36 (46.7)	18 (56.2)	18 (40.0)	0.16
Hypertension	35 (45.4)	15 (46.9)	20 (44.4)	0.83
Chronic artery disease	7 (9.1)	4 (12.5)	3 (6.7)	0.44
Chronic kidney disease	2 (2.6)	1 (3.1)	1 (2.2)	1.00
Time from symptoms to ICU admission, day	5 (4–7)	5 (4–11)	5 (4–7)	0.21
Time from ICU admission to intubation, day	0 (0–1)	0 (0–2)	0 (0–1)	0.50
Time from symptoms to intubation, day	5 (3–10)	6 (3–11)	5 (3–9)	0.46
Vital signs on ICU admission				
Temperature (max) ≥ 38 °C, *n* (%)	28 (36.4)	11 (34.4)	17 (37.8)	0.76
Heart rate (max), beats·min^−1^	105 ± 20	105 ± 21	105 ± 20	0.99
Respiratory rate (max), breaths·min^−1^	32 ± 8	30 ± 8	32 ± 8	0.30
Laboratory data on ICU admission				
C–reactive protein, mg·L^−1^	159 (68–244)	104 (58–166)	198 (99–295)	0.003
Leucocytes count, × 10^9^·L^−^^1^	9.8 (7.6–13.0)	10.6 (7.4–14.3)	9.7 (7.7–12.2)	0.53
Lymphocytes count, × 10^9^·L^−^^1^	0.79 (0.49–1.00)	0.88 (0.54–1.06)	0.71 (0.47–0.99)	0.22
Lymphocytes ≤ 1 × 10^9^·L^−^^1^; *n* (%)	58 (75.3)	23 (71.9)	35 (77.8)	0.55
Platelet count, × 10^9^·L^−^^1^	251 (183–320)	245 (160–319)	256 (197–320)	0.51
Procalcitonin, ng·mL^−1^	0.60 (0.21–3.71)	0.44 (0.21–2.07)	0.96 (0.23–7.58)	0.11
International normalized ratio	1.2 (1.1–1.3)	1.2 (1.1–1.3)	1.2 (1.1–1.4)	0.18
Activated partial thromboplastin time; s	34.1 (30.0–38.3)	33.2 (28.1–37.7)	34.6 (31.2–38.4)	0.25
D–dimer, µg·mL^−1^ (normal reference: <0.05)	3.7 (1.7–4.0)	3.4 (1.3–4.0)	4.0 (1.9–4.0)	0.35
D–dimer ≥ 2 µg·mL^−^^1^, *n* (%)	56 (72.7%)	23 (71.9)	33 (73.3)	0.89
Fibrinogen, g·L^−1^	6.1 (4.6–7.2)	5.8 (4.1–6.7)	6.3 (5.2–7.7)	0.21
Ferritin, µg·L^−1^ (reference range: 36–480)	1561 (895–2484)	1854 (968–2479)	1406 (748–2582)	0.32
Interleukin 6, ng·L^−1^	279 (130–1130)	174 (103–466)	665 (176–2438)	0.005
Alanine aminotransferase, IU·mL^−^^1^	37 (27–65)	37 (30–66)	38 (25–63)	0.67
Aspartate aminotransferase, IU·mL^−^^1^	55 (36–91)	50 (36–84)	59 (36–95)	0.28
Total bilirubin, µmol·L^−1^	11.6 (8.1–18.2)	10.5 (8.1–15.8)	12.0 (8.3–20.4)	0.55
Creatinine, µmol·L^−1^	83 (65–154)	74 (63–123)	91 (65–155)	0.19
Albumin, g·L^−1^	30 (27–34)	31 (27–34)	29 (27–33)	0.39

SOFA, Sequential Organ Failure Assessment. SAPS, Simplified Acute Physiology Score. ICU, intensive care unit. Data are shown as mean ± SD, median [1st–3rd quartile], and count (%). *p* < 0.05 was considered statistically significant.

**Table 2 jcm-10-00760-t002:** Comparisons of baseline oxygenation and ventilator variables, and treatments during the ICU stay, between Methylprednisolone and non-Methylprednisolone groups.

Variables	All Patients (*n* = 77)	Methylprednisolone (*n* = 32)	Non-Methylprednisolone (*n* = 45)	*p*-Value
Oxygenation variables on ICU admission				
PaO_2_, mmHg	68 (57–83)	70 (61–84)	67 (56–81)	0.71
PaCO_2_, mmHg	44 (34–56)	46 (33–54)	43 (34–58)	0.74
FiO_2_	0.9 (0.6–1)	0.8 (0.6–1)	1 (0.6–1)	0.43
PaO_2_/FiO_2_ ratio, mmHg	83 (62–131)	85 (67–134)	83 (61–128)	0.83
SaO_2_, %	92 (88–94)	93 (88–94)	91 (88–93)	0.33
Lactate levels, mmol·L^−1^	1.45 (1.20–1.90)	1.40 (1.20–1.90)	1.45 (1.20–1.90)	0.83
Treatments during the ICU stay, *n* (%)				
Vasopressor support	65 (84.2)	23 (71.9)	42 (93.3)	0.01
Renal replacement therapy	24 (31.2)	8 (25.0)	16 (35.6)	0.32
Prone position	50 (64.9)	22 (68.7)	28 (62.2)	0.55
Neuromuscular blocker agents	63 (81.2)	26 (81.2)	37 (82.2)	1.00
Tocilizumab	67 (87.0)	26 (81.2)	41 (91.1)	0.30
Hydroxychloroquine	30 (39.0)	12 (37.5)	18 (40.0)	0.82
Favipiravir	23 (29.9)	11 (34.4)	12 (26.7)	0.47
Lopinavir/ritonavir	18 (23.4)	8 (25.0)	10 (22.2)	0.78
Ventilator parameters on ICU admission				
Tidal volume, mL·kg^−1^ IBW	6.5 (6.0–7.0)	6.5 (6.1–7.0)	6.5 (5.7–7.2)	0.65
Plateau pressure, cmH_2_O	28 (26–30)	28 (26–30)	28 (27–30)	0.84
Positive end expiratory pressure, cmH_2_O	12 (10–14)	12 (10–14)	12 (10–14)	0.58
Driving pressure, cmH_2_O	16 (13–18)	16 (13–19)	17 (14–18)	0.59
Static compliance, mL.cmH_2_O^−1^	27.0 (22.0–34.1)	28.6 (22.5–35.4)	25.0 (21.1–32.5)	0.21

PaO_2_, arterial oxygen tension. PaCO_2_, arterial CO_2_ tension. FiO_2_; Inspiratory oxygen fraction. SaO_2_, arterial oxygen saturation. IBW, ideal body weight. ICU, intensive care unit. Data are shown median [1st–3rd quartile], and count (%). *p* < 0.05 was considered statistically significant.

**Table 3 jcm-10-00760-t003:** Multivariable competing-risks regression analysis.

Variables	SHR	95% Confidence Interval	*p*-Value
Methylprednisolone treatment (refer: no)	0.10	0.02–0.45	0.003
Age, year	1.04	0.95–1.14	0.42
Body mass index, kg·m^−2^	1.16	1.01–1.33	0.03
SOFA score	0.92	0.75–1.13	0.44
Lactate, mmol·L^−1^	3.93	1.78–8.70	0.001
PaO_2_/FiO_2_ ratio, mmHg	1.00	0.98–1.02	0.98
SaO_2_, %	1.09	0.88–1.36	0.41
D-dimer, µg·mL^−1^	0.54	0.31–0.94	0.03
Procalcitonin, ng·mL^−1^	1.02	1.002–1.046	0.03
Interleukin 6, ng·L^−1^	1.00	0.99–1.00	0.98
C-reactive protein, mg·L^−1^	1.00	0.99–1.00	0.49
Time from symptoms onset to Methylprednisolone, day	1.02	0.78–1.34	0.86
Creatinine, µmol·L^−1^	1.00	0.99–1.00	0.22
Vasopressor support, (refer: no)	4.33	0.47–40.15	0.20
Renal replacement therapy, (refer: no)	1.90	0.51–7.10	0.34
Prone position, (refer: no)	16.70	1.70–260.62	0.04
Neuromuscular blocker agents, (refer: no)	2.60	0.12–56.41	0.54
Tocilizumab, (refer: no)	0.20	0.02–2.43	0.21
Comorbidities, (refer: no)	2.35	0.71–7.74	0.16
Tidal volume, mL·kg^−1^ IBW	0.99	0.54–1.79	0.97
Plateau pressure, cmH_2_O	0.97	0.70–1.34	0.86
Positive end expiratory pressure, cmH_2_O	0.74	0.49–1.11	0.15
Driving pressure, cmH_2_O	0.91	0.72–1.16	0.46

SOFA, Sequential Organ Failure Assessment. PaO_2_, arterial oxygen tension. FiO_2_, Inspiratory oxygen fraction. SaO_2_, arterial oxygen saturation. IBW, ideal body weight. SHR, sub hazards ratio. *p* < 0.05 was considered statistically significant.

## Data Availability

The data presented in this study are available on request from the corresponding author. The data are not publicly available due to the Ethics Committee restrictions.

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
