# Peer review of "Effects of Methylprednisolone on Ventilator-Free Days in Mechanically Ventilated Patients with Acute Respiratory Distress Syndrome and COVID-19: A Retrospective Study"

_jcm, 2021, doi:10.3390/jcm10040760_

Round 1

Reviewer 1 Report

Major comments:

1) The basis for using steroids has been the RECOVERY trial, where dexamethasone was used. What was the reason to use methylprednisolone instead?

2) Similarly, why as 1 mg/kg used? This would be higher than 6 mg of dexamethasone. Duration usage is shorter than the recommended 10 days.

3) The majority of the patients were male. Do you think this could impact the results? I would also add that as a limitation to the study.

4) Can the authors speculate why the steroids group required less vasopressor support?

5) Concerning the incidence of ‘positive’ blood cultures, what type of organisms were identified? It is surprising that a short course of steroids would lead to that. I doubt this can be attributed to toclizumab since the majority of  the methylpred and non methylpred groups received it (81.2 and 91.1%).

I believe there is a typo line 223 (81.2 not 91.1% received methylpred according to table 2)

Minor comments:

Edit line 61 ‘patients with severe COVID-19 severe present nonspecific hyperinflammatory responses’. Severe is  redundant

Edit line 71  ‘were not reported in these patients yet are associated with outcome’. Maybe use but or consider rewriting that sentence.  

Author Response

Reviewer#1

Reviewer: Major comments: 1) The basis for using steroids has been the RECOVERY trial, where dexamethasone was used. What was the reason to use methylprednisolone instead?

Response: We thank the reviewer for this important comment. We started using methylprednisolone before the publication of the RECOVERY trial, which was published on July 17th, 2020. We included patients between March 1st and May 29th, so before the results of the RECOVERY trial. We hope that we have addressed the reviewer’s concern appropriately.

Reviewer: 2) Similarly, why as 1 mg/kg used? This would be higher than 6 mg of dexamethasone. Duration usage is shorter than the recommended 10 days.

Response: We thank the reviewer for this point. As we explained above, the RECOVERY trial was published after the period of inclusion of our study. The choice and dose of steroids were at the discretion of the treating Intensivist. We hope that we have addressed the reviewer’s concern appropriately.

Reviewer: 3) The majority of the patients were male. Do you think this could impact the results? I would also add that as a limitation to the study.

Response: We thank the reviewer for this important comment. We do not think that a very high proportion of male patients (93.5%) would have impacted the results. Indeed, the distribution of male gender was well balanced between the methylprednisolone and non-methylprednisolone groups (93.7% vs. 93.3%, p=1.00). The CoDEX trial included 60% male patients and 40% female patients, and they had the same results. In the meta-analysis published in JAMA regarding steroids in hospitalized patients (JAMA. 2020;324(13):1330-1341), the association between steroids and outcome was not different in male and female patients.  We hope that we have addressed the reviewer’s concern appropriately.

Reviewer: 4) Can the authors speculate why the steroids group required less vasopressor support?

Response: We thank the reviewer for his/her comment. It is hard to speculate on that. It could be a selection bias since our study was retrospective. It could also be that methylprednisolone had beneficial effects on the blood vessels resulted in decreasing sepsis-induced vasodilatation. Whatever was the reason, we adjusted for vasopressors, so the impact of methylprednisolone was independent of vasopressors. We hope that we have addressed the reviewer’s concern appropriately.

Reviewer: 5) Concerning the incidence of ‘positive’ blood cultures, what type of organisms were identified? It is surprising that a short course of steroids would lead to that. I doubt this can be attributed to toclizumab since the majority of the methylpred and non methylpred groups received it (81.2 and 91.1%).

Response: We thank the authors for this important comment. The organisms identified in the blood cultures were gram-positive bacilli (52.6%), gram-negative bacilli (47.4%), and fungemia (10.5%). The sum is more than 100% as some blood cultures grew more than one organism. We agree with the reviewer that the high incidence of positive blood culture cannot be attributed to the use of tocilizumab alone. This could be due to the combination of tocilizumab and methylprednisolone (2 immunosuppressive treatments), as the group with no methylprednisolone received tocilizumab in 91% of cases and had a lower rate of blood cultures. However, this explanation is only speculative as our study cannot address this issue. More studies are needed to address the incidence of positive blood cultures in patients who received both tocilizumab and steroids. We hope that we have addressed the reviewer’s concern appropriately.

Reviewer: I believe there is a typo line 223 (81.2 not 91.1% received methylpred according to table 2)

Response: The typo is in the text, not in table 2. The percentages of tocilizumab in table 2 are correct. We changed 91% to 81% on page 7, line 223, and page 9, line 293. We apologize for the typo.

Reviewer: Minor comments: Edit line 61 ‘patients with severe COVID-19 severe present nonspecific hyperinflammatory responses’. Severe is  redundant

Response: We thank the reviewer for the thorough reading of the manuscript. We have now removed “severe” after COVID-19, page 2, line 61.

Reviewer: Edit line 71  ‘were not reported in these patients yet are associated with outcome’. Maybe use but or consider rewriting that sentence.  

Response: We thank the reviewer for the thorough reading of the manuscript. The sentence has now been changed to: “were not reported in these patients but are associated with outcome” (line 71) as suggested by the reviewer.

We thank the reviewer for his/her comments that have helped improve the manuscript's quality.

Reviewer 2 Report

Thank you for the invitation to review this manuscript examining outcomes in COVID-19 patients who did and did not receive methylprednisolone during the course of their critical illness.  The paper addresses an important issue and builds on the existing literature regarding corticosteroids in severe COVID-19.  The authors show a significant association between methylprednisolone administration and alive ventilator free days at 28 days, even after controlling for numerous potential confounding factors.

Overall the work appears robust, with appropriate statistical methods and close attention to detail.  The manuscript is well written and generally easy to follow.

As a retrospective study, however, there are some substantial limitations and these are perhaps downplayed by the authors.  Causation cannot be inferred from these findings, regardless of the statistical approaches used, and it is important that the authors emphasise this more clearly.  The title should be changed to avoid any suggestion that this is a clinical trial.  The reference to this limitation should be expanded within the discussion section.

Further areas requiring amendment re: causation: P9, L303-304: it is not appropriate to state that methylpred increased VFDs by 4.6 days; as well as the issue of implying causation this overstates the study findings by using the unadjusted values.  Line 319 on P9 also implies causation regarding hospital length of stay - this should be rephrased. P1, L22 (study aim) should also be rephrased. 

I would also welcome more information on the criteria used in the decision to administer methylprednisolone in this patient population (and indeed tocilizumab).  Although the two groups appear well-matched, there must have been a factor(s) that influenced the choice to give corticosteroids; it is possible that this could influence patient outcomes and this must be considered in the analysis and/or discussion.  Were the non-methylprednisolone group given any other corticosteroids? 

I am also surprised by the sex mix of the population under study - 94% male is a very high proportion.  Do the authors have a comment to explain this?

I would welcome some commentary on the reasons for selection of methylprednisolone rather than dexamethasone - the differing pharmacological profiles should be discussed.

In summary, the findings are valuable but the interpretation should be phrased more appropriately and cautiously to ensure the implications of the study are not overstated.

Minor comments:

P2, L61: typo, "severe COVID-19 severe"

P3, L105: remove full stop and merge sentences "For patients who required mechanical ventilation for more than 28 days. The number of VFDs was 0"

P3, L133: APTT should be capitalised 

Please correct the use of "p≤0.05" to "p<0.05" in results tables as the threshold for statistical significance.

P9, L284: typo - "trail" should be corrected to "trial"

P9, L296: the word "not" is missing

Author Response

Reviewer#2

Reviewer: Overall the work appears robust, with appropriate statistical methods and close attention to detail.  The manuscript is well written and generally easy to follow.

Response: We thank the reviewer for the supportive comment.

Reviewer: As a retrospective study, however, there are some substantial limitations and these are perhaps downplayed by the authors.  Causation cannot be inferred from these findings, regardless of the statistical approaches used, and it is important that the authors emphasise this more clearly.

Response: We thank the reviewer for the important point. We fully agree with the reviewer’s comment that the retrospective design is a major limitation of the study, and even by using advanced statistical methods and adjusting for many confounders, selection bias cannot be ruled out, and causation cannot be inferred from our findings. We already highlighted that in the limitation of the study, page 9, lines 310-312: “Second, despite multivariable analysis and adjustment for potential confounders, we cannot rule out bias selection or residual confounding.” We have now emphasized this issue more in the revised manuscript. We hope that we have addressed the reviewer’s concern appropriately.

Reviewer: The title should be changed to avoid any suggestion that this is a clinical trial. 

Response: We thank the reviewer for this comment. We have now changed the title to: “Effects of methylprednisolone on ventilator-free days in mechanically ventilated patients with acute respiratory distress syndrome and COVID-19: A retrospective study.

Reviewer: The reference to this limitation should be expanded within the discussion section.

Response: We thank the reviewer for this comment. We have now highlighted this issue more in the discussion. We have now made some changes to the discussion: page 9, lines 304-306: “Methylprednisolone was associated with increases in the number of days alive and free from mechanical ventilation by 4.6 days and shortening hospital length of stay.” We also added, page 9, line 312: “Thus, no causation can be inferred from our findings.” We hope that we have addressed the reviewer’s concern appropriately.

Reviewer: Further areas requiring amendment re: causation: P9, L303-304: it is not appropriate to state that methylpred increased VFDs by 4.6 days; as well as the issue of implying causation this overstates the study findings by using the unadjusted values.

Response: We thank the reviewer for this comment. We have now rephrased the sentence, page 9, lines 304-306, as suggested by the reviewer: “Methylprednisolone was associated with increases in the number of days alive and free from mechanical ventilation by 4.6 days and shortening hospital length of stay.” We hope that we have addressed the reviewer’s concern appropriately.

Reviewer: Line 319 on P9 also implies causation regarding hospital length of stay - this should be rephrased.

Response: We respectfully disagree with the reviewer’s comment. Indeed, the sentence, page 9, line 319: “Methylprednisolone was independently associated with the increase in the number of days alive and free of mechanical ventilation over 28 days in mechanically ventilated ARDS patients with COVID-19” does not imply causation but an association between methylprednisolone and VFDs. We did not write methylprednisolone increased VFDs, but we clearly mentioned that it was independently associated with VFDs. This conclusion is supported by our results. By performing a multivariable analysis and after adjustment for potential confounders, we found an independent ASSOCIATION (not causation) between methylprednisolone and VFDs (Table 3).  From a methodological and statistical stand point, this sentence is correct. We hope that we have addressed the reviewer’s concern appropriately.

Reviewer: P1, L22 (study aim) should also be rephrased. 

Response: We thank the reviewer for this comment. We have now rephrased the sentence, page 1, line 22, to: “We aimed to determine whether methylprednisolone is associated with increases in the number of ventilator-free days (VFDs) among these patients.” as suggested by the reviewer. We hope that we have addressed the reviewer’s concern appropriately.

Reviewer: I would also welcome more information on the criteria used in the decision to administer methylprednisolone in this patient population (and indeed tocilizumab). 

Response: We thank the reviewer for this important point. We did not have clear criteria for using methylprednisolone or tocilizumab. As stated in the manuscript, their uses were at the discretion of the treating physician. Regarding methylprednisolone, the Intensivist in charge of the patient decided to give it or not based on his assessment as there was no evidence about the efficacy of steroids at that time. The RECOVERY study was not published yet. Some intensivists from our team were a believer in steroids, and some others were not.  Regarding tocilizumab, its administration's decision was based on a discussion between the infectious disease physician and the Intensivist treating the patient. We hope that we have addressed the reviewer's concern appropriately.

Reviewer: Although the two groups appear well-matched, there must have been a factor(s) that influenced the choice to give corticosteroids; it is possible that this could influence patient outcomes and this must be considered in the analysis and/or discussion.

Response: We thank the reviewer for this important point. We agree with the reviewer’s comment that the choice of giving steroids could influence patient outcomes. However, we tried as much as possible to overcome this issue by using multivariable analysis and adjusting for potential confounders. As the reviewer rightly stated, our two groups were well-matched. We understand that we could not adjust for unmeasured confounders, but this limitation, which we clearly highlighted in the discussion, is related to all non-randomized clinical trials. We already stated in the discussion, page 9, lines 310-312: “Second, despite multivariable analysis and adjustment for potential confounders, we cannot rule out bias selection or residual confounding.” We meant by selection bias “the choice to give steroid or not to the patient.” We hope that we have addressed the reviewer’s concern appropriately.

Reviewer: Were the non-methylprednisolone group given any other corticosteroids? 

Response: We thank the reviewer for this important point. No, the non-methylprednisolone group did not receive other steroids.

Reviewer: I am also surprised by the sex mix of the population under study - 94% male is a very high proportion.  Do the authors have a comment to explain this?

Response: We thank the reviewer for this important point. In our hospital, we received many expatriates’ workers from India, Pakistan, the Philippines, etc. that were predominantly male.

Reviewer: I would welcome some commentary on the reasons for selection of methylprednisolone rather than dexamethasone - the differing pharmacological profiles should be discussed.

Response: Thank you for this point. At the hat time, the RECOVERY trial was not yet published. When we usually use steroids in non-COVID-19 ARDS patients, we use methylprednisolone. That is why we chose methylprednisolone. We added in the introduction, page 2, lines 80-81, a brief description of the differences of pharmacological profiles between dexamethasone and methylprednisolone: “Methylprednisolone has lower potent anti-inflammatory effects and shorter plasma half-time than dexamethasone.”  We hope that we have addressed the reviewer’s concern appropriately.

Reviewer: In summary, the findings are valuable but the interpretation should be phrased more appropriately and cautiously to ensure the implications of the study are not overstated.

Response: We thank the reviewer for the supportive comment. We hope that we have addressed all the reviewer’s concerns appropriately.

Reviewer: P2, L61: typo, "severe COVID-19 severe"

Response: Thank you for your thorough reading of the manuscript. We have now corrected the typo in the main text.

Reviewer: P3, L105: remove full stop and merge sentences "For patients who required mechanical ventilation for more than 28 days. The number of VFDs was 0"

Response:  Thank you for your thorough reading of the manuscript.  We have now changed the sentence as suggested by the reviewer.

Reviewer: P3, L133: APTT should be capitalised 

Response: Thank you for your thorough reading of the manuscript. We have now capitalized APTT in the main text.

Reviewer: Please correct the use of "p≤0.05" to "p<0.05" in results tables as the threshold for statistical significance.

Response: Thank you for your thorough reading of the manuscript. We have now made the changes in tables as requested by the reviewer.

Reviewer: P9, L284: typo - "trail" should be corrected to "trial"

Response: Thank you for your thorough reading of the manuscript.  We have now corrected the typo in the main text.

Reviewer: P9, L296: the word "not" is missing

Response: Thank you for your thorough reading of the manuscript. We have now corrected the typo in the main text.

We thank the reviewer for his/her comments that have helped improve the manuscript's quality.

Reviewer 3 Report

The study of Badr et al. touches on an important and very hot topic about the role of systemic steroids in the treatment of COVID19. Its results confirms the rationale of administration of steroids in patients with severe course of the disease especially in those requiring mechanical ventilation. As such it constitutes another argument pro steroid treatment. I cannot see any  methodological drawbacks. All weak points were listed by authors and everyone has to agree with them. The retrospective design of the study, small sample and confounders, like tocilizumab given mainly to subjects treated with methylprednisolone, make the conclusions less evident.

Author Response

Reviewer#3

Reviewer: The study of Badr et al. touches on an important and very hot topic about the role of systemic steroids in the treatment of COVID19. Its results confirm the rationale of administration of steroids in patients with severe course of the disease especially in those requiring mechanical ventilation. As such it constitutes another argument pro steroid treatment. I cannot see any methodological drawbacks. All weak points were listed by authors and everyone has to agree with them. The retrospective design of the study, small sample and confounders, like tocilizumab given mainly to subjects treated with methylprednisolone, make the conclusions less evident.

Response: We thank the reviewer for the supportive comment.  

Reviewer: The retrospective design of the study, small sample and confounders, like tocilizumab given mainly to subjects treated with methylprednisolone, make the conclusions less evident.

Response: We thank the reviewer for this point. We agree with the reviewer's comment. However, we adjust for tocilizumab in the multivariable analysis to minimize tocilizumab's effects on the relationship between methylprednisolone and VFDs. We hope that we have addressed all the reviewer's concerns appropriately.

We thank the reviewer for his/her comments that have helped improve the manuscript's quality.